# Super-Resolution Imaging Reveals Dynamic Reticular Cytoophidia

**DOI:** 10.3390/ijms231911698

**Published:** 2022-10-02

**Authors:** Yi-Fan Fang, Yi-Lan Li, Xiao-Ming Li, Ji-Long Liu

**Affiliations:** 1School of Life Science and Technology, Shanghai Tech University, Shanghai 201210, China; 2Department of Physiology, Anatomy and Genetics, University of Oxford, Oxford OX1 3PT, UK

**Keywords:** CTP synthase, cytoophidium, fluorescence recovery after photobleaching (FRAP), stimulated emission depletion (STED)

## Abstract

CTP synthase (CTPS) can form filamentous structures termed cytoophidia in cells in all three domains of life. In order to study the mesoscale structure of cytoophidia, we perform fluorescence recovery after photobleaching (FRAP) and stimulated emission depletion (STED) microscopy in human cells. By using an EGFP dimeric tag as a tool to explore the physical properties of cytoophidia, we find that cytoophidia are dynamic and reticular. The reticular structure of CTPS cytoophidia may provide space for other components, such as IMPDH. In addition, we observe CTPS granules with tentacles.

## 1. Introduction

In addition to organelles with membranes, proteins with important functions in the cell can also be compartmented into membraneless organelles. CTP synthase (CTPS), a metabolic enzyme for de novo synthesis of CTP, was found to form filament-like compartments in cells called cytoophidia [1]. Describing their shape vividly, the word “cytoophidia” means “cellular snakes” in Greek. Cytoophidia were found in many species in all three domains of life, which means cytoophidia are conserved in evolution [1,2,3,4,5,6,7,8,9,10,11,12,13,14,15,16,17,18,19].

A glutamine analog, 6-diazo-5-oxo-L-norleucine (DON), promotes cytoophidia formation in *Drosophila* and human cells [5]. DON binds CTPS with covalent bonds [20]. Glutamine deprivation promotes cytoophidium formation in mammalian cells [21]. IMPDH can form cytoophidia [22] and both CTPS and IMPDH are related to glutamine and NH_3_ metabolism. It was reported that there is an interaction between CTPS and IMPDH [23]. The function of cytoophidia may be closely related to glutamine and NH_3_ metabolism.

In metabolic regulation, the activity of CTPS is inhibited via filament formation [24,25]. The half-life of CTPS is prolonged when forming cytoophidia [26]. Given the high-level metabolism in cancer cells, cytoophidium formation is highly related to oncogenes. Myc is required for cytoophidia assembly, and cytoophidia formation is regulated by Myc expression levels [27]. Ack kinase regulates cytoophidium morphology and CTPS activity [28]. Cytoophidium assembly was found to be regulated by the mTOR-S6K1 pathway [29]. Cytoophidia were also found in human hepatocellular carcinoma [10].

CTPS can be assembled into thin filaments in vitro, and the structures of CTPS filaments at near-atomic resolution have been solved by cryo-EM [20,25,30]. However, nanometer-scale CTPS filaments are different from the micron-scale cytoophidia observed via confocal microscopy. How CTPS filaments assemble into big micron-scale cytoophidia is still unclear. The physical properties of cytoophidia at the mesoscale remain to be explored.

To study cytoophidium properties in human cell lines, we performed fluorescence recovery after photobleaching (FRAP) microscopy to study the dynamic characteristics and stimulated emission depletion (STED) microscopy to study the super-resolution structure. By measuring the intensity and recovery speed of bleached ROIs, we were able to quantify the relative dynamic characteristics of cytoophidia under different treatments. STED allows fluorescence imaging to achieve a resolution of 50 to 70 nm [31,32].

## 2. Results

### 2.1. Assembly of CTPS Filaments into Cytoophidia

The cytoophidium is a compartment of metabolic enzymes, such as filamentous CTPS, which can be observed via confocal microscopy [1,2,3]. In vitro experiments showed that CTPS can also form filaments built from tetramer units [20,25,30]. In human cells, CTPS can be assembled into cytoophidia under DON treatment or glutamine deprivation. In this study, we did not distinguish hCTPS1 and hCTPS2. We found that after DON treatment, hCTPS1 can form granules in 293T cells with hCTPS1 over-expression (Figure 1A,B). When we observed living cells, CTPS granules existed in a small population of cells. CTPS granules can exist in the same cells as cytoophidia (Figure 1A,B).

How CTPS filaments are arranged in cytoophidia remains unclear. To conceive the arrangement model from CTPS filaments to cytoophidia, we constructed hCTPS1-overexpression vectors with different fluorescence proteins and different mutations (Appendix A). In order to show and evaluate the effect of exogenous protein overexpression in the experiment, we compared the protein levels of overexpressed hCTPS1 and endogenous hCTPS1/2 (Figure 1D) and tested the transfection efficiencies (Figure 1E,F). In transfection-positive cells, exogenous hCTPS1 expression was approximately twice as high as that of endogenous hCTPS1/2 (Figure 1G).

EGFP is a weak dimer while EGFP^A206K^ is a monomer [33]. Overexpression of *hCTPS1-EGFP* forms cytoophidium-like condensates in 293T cells (Figure 1H). The force of forming dimer between EGFP pulls hCTPS1 together, and hCTPS is assembled into filaments. The hCTPS filaments may be compressed together by a simple force of EGFP dimerization (Figure 1C). The hCTPS1-EGFP group was a control for cytoophidium induction. CTPS1 with the H355A mutation disassembles cytoophidia. We found that overexpressed *hCTPS1^H355A^-EGFP* could not form cytoophidia in 293T cells (Figure 1H).

### 2.2. Dynamic Equilibria of Cytoophidia

To test the dynamic characteristics of cytoophidia, we performed FRAP on four groups of hCTPS1 cytoophidia (Figure 2A,B). The intensity of the FRAP ROI was normalized as $Normalized\;Intensity=\frac{I{\left(non-bleach\;ROI\right)}_{Pre-bleach,\;0}-I{\left(background\right)}_{0}\;}{I{\left(non-bleach\;ROI\right)}_{n}-I{\left(background\right)}_{n}}\;\times \left[I{\left(bleach\;ROI\right)}_{n}-I{\left(background\right)}_{n}\right]$ [34]. The bleached ROIs on cytoophidia induced by 20 μg/mL for 8 h (low concentration and short time) before imaging recovered very quickly (Figure 2C). However, ROIs in cells with *hCTPS1* overexpression treated with DON in 100 μg/mL (higher concentration) recovered fluorescence slowly and ended at a lower intensity (Figure 2D; Appendix A).

By extending the DON treatment time to 25 h, ROIs were able to recover as quickly as possible and in a relatively short time (Figure 2E; Appendix A). The fluorescence intensity restored by bleaching the ROIs of cytoophidium-like condensates of hCTPS-EGFP cells was very low (Figure 2F; Appendix A). There was a significant difference in dynamics between hCTPS1-EGFP cytoophidium-like condensates and DON-induced hCTPS cytoophidia at low concentrations and over short time periods (Figure 2G).

This shows that DON-induced cytoophidia have very different dynamic characteristics from hCTPS-EGFP cytoophidium-like condensates. DON-induced cytoophidia seem not to be assembled by simple forces, like the hCTPS-EGFP cytoophidium-like condensates.

The bleached ROIs gradually recovered throughout, rather than from any particular side. Neither of the two ROIs moved to either side, nor were they far away or close to each other (Figure 2C). Our results showed that bleached hCTPS1 molecules in cytoophidia could exchange with free hCTPS1 molecules in the cytosol (Figure 2H).

### 2.3. The Reticular Structure of the hCTPS1 Cytoophidium and Its Localization with hIMPDH2

In order to build a model to fit the dynamic-equilibrium characteristics of cytoophidia, we obtained super-resolution structures of hCTPS cytoophidia using stimulated emission depletion microscopy (STED). The images under conventional confocal microscopy could not show the structure inside cytoophidia, while the STED images revealed the super-resolution structure with a resolution of 50 to 70 nm (Figure 3A and Appendix A), which implied a possible mechanism of highly dynamic cytoophidia under FRAP. We estimated the resolution by measuring the distance between two distinguishable nearby particles and it was 50 to 70 nm (Appendix A).

STED revealed a heterogeneous structure for hCTPS cytoophidia. Some parts were relatively more condensed, while other parts were looser. More importantly, it seemed that there were many tiny filaments inside, which in different orientations formed reticular structures (Figure 3A; Appendix A).

However, the super-resolution results were homogeneous inside hCTPS1-EGFP cytoophidium-like condensates (Figure 3A). The structures of the hCTPS cytoophidia and those of the hCTPS1-EGFP cytoophidia were totally different. No condensed and loose parts or reticular structures knitted with tiny filaments could be observed in the cytoophidium-like condensates.

When performing super-resolution imaging, there might be some interfering factors against reliability of the super-resolution imaging results, such as the efficiencies of antibodies, optical properties of the fluorescent labeling, the steric hindrance of fluorescence proteins, and the influence of sample preparation. To eliminate the effects of antibodies and Cy5, we used 293T cells with hCTPS1-miRFP670nano for the live-cell imaging. We performed immunofluorescence staining on DON-treated 293T cells to eliminate the effects of potential steric hindrance of fluorescence protein tags and overexpression. Both results showed reticular structures (Figure 3A). To avoid the difference in optical properties between Cy5 and EGFP, we also performed immunofluorescence staining with Cy5 on hCTPS1-EGFP-overexpressed 293T cells, and the signal obtained was from Cy5 (Figure 3A).

In addition, we performed immunofluorescence staining on SW480 cells cultured in glutamine-free medium, which showed a reticular structure (Figure 3A). Glutamine is an NH_3_ donor in metabolic reactions. This means that the reticular structure of hCTPS is not only a phenomenon induced by DON but also a common structure of metabolic enzymes when cells are under metabolic stress. Without changing the super-resolution structural results, deconvolution of the STED images could improve their resolution and signal-to-noise ratios (Figure 3A; Appendix A).

These tiny filaments appeared as subunits of hCTPS cytoophidia (Figure 3A,B). In vitro experiments showed that CTPS can be assembled into filaments [20,25,30]. Based on the in vitro and in vivo results, we envisioned a model to illustrate the reticular structure of hCTPS cytoophidia (Figure 3B). Inside cytoophidia, subunit filaments are weaved into a reticulation. The model can make FRAP results clearer. The dynamic equilibrium of assembly and disassembly occurs in the tiny filaments of hCTPS, rather than the assembly and disassembly of the whole cytoophidium. FRAP procedures performed on untreated and dispersive hCTPS1 signals resulted in fast recovery, which meant that we were unable to capture the images after bleaching, which were similar to the images before bleaching (Appendix A). Due to the limitation of STED resolution, it is unclear whether the subunit filament is one CTPS filament, a bundle of CTPS filaments or some other form of CTPS.

After obtaining the super-resolution reticular structures, we wanted to determine the reason for and function of this reticular structure. There might be some unknown molecules in the space between hCTPS filaments (Figure 3B). It was reported that IMPDH2 interacted with CTPS1 cytoophidia under DON treatment [23]. The cytoophidia of hCTPS1 and hCTPS2 were located together in 293T cells (Appendix A). IMPDH and CTPS are both part of the glutamine and NH_3_ metabolic pathways (Appendix A). We overexpressed *hCTPS1-EGFP^A206K^*, a monomer version of EGFP, and labeled IMPDH2 with Cy5 by immunofluorescence staining. Under DON treatment, IMPDH2 and hCTPS1 were localized spatially adjacent to each other (Figure 3C). IMPDH2 and hCTPS1 were not exclusive but were positioned mutually in each Z stack (Figure 3D). Therefore, IMPDH2 could be one of the molecules located between hCTPS1 filaments.

### 2.4. CTPS Granules with Tentacles

We performed live-cell imaging via confocal microscopy in 293T cells overexpressing *hCTPS1-mCherry* and found that hCTPS1 could not only form cytoophidia but also formed DON-induced granules (Figure 1A). The movement of most granules was a random walk, like that of ordinary granules in cells (Figure 1B). Fortunately, when we observed live cells treated with DON, we found a filiform structure connecting hCTPS1 granules (Figure 4A). We name these filiform structures connecting granules “tentacles”, and the main part of the granules is called the “granule body”. The tentacle slowly extended out of the granule body and retracted as quickly as a rubber band after reaching its longest length (Figure 4B).

We wanted to know the function of the granular tentacles. We found that tentacles were different from small granule bodies. When the tentacle stretched out, the granule body moved from one side of the tentacle to the other side along the tentacle, and then the tentacle retracted into the granule body in the new location (Figure 4C). Granules with tentacles move with clear direction along the tentacles rather than in a random walk. The movement of tentacled granules was different from that of non-tentacled granules. Granular tentacles are tiny structures that bridge granules, move granules and retract after extension (Figure 4D).

## 3. Discussion

Taking advantage of multiple fluorescence tags, we study the physical characteristics of hCTPS1-containing compartments via fluorescence microscopy. We perform FRAP and STED analyses to reveal the dynamic and reticular structure of cytoophidia. In addition, we observe that hCTPS1 forms granules with tentacles.

### 3.1. Cytoophidia Are Not Condensates

Protein compartments, similar to droplets, can be assembled by physical forces in cells [35]. Since the discovery of CTPS-forming cytoophidia, the exact phases of cytoophidia and the arrangements of CTPS in cytoophidia are still unclear. Cytoophidia are presumed to be static bundles of filaments [4](Liu, 2011) or in a liquid phase, just like LLPS. However, when we performed live-cell imaging on CTPS cytoophidia, we found that CTPS can not only form long filamentous structures, that is, cytoophidia, but also form granules in the same cell (Figure 1A). This means that, as compartments of CTPS, cytoophidia may not be static, concentrated and rigid structures. For another hypothesis, the puzzling question is why this compartment is not a spherical droplet if it is in the liquid phase, such as LLPS.

According to previous studies, the residue 355H of CTPS (CTPS-355H) is the key site for the formation of this filamentous structure. CTPS-355H lying at the tetramer–tetramer interface plays a critical role in CTPS polymerization. In in vitro experiments, the CTPS tetramer assembly mechanism of cytoophidia is more like that of actin filaments than droplets in cells assembled by physical force [36].

To study the role of CTPS-355H, we used both dimeric EGFP and monomeric EGFP^A206K^ tags [32]. We generated *hCTPS1^H355A^* mutations. hCTPS1-EGFP^A206K^ can form cytoophidia with DON treatment. Without DON treatment, hCTPS1-EGFP^A206K^ cannot form cytoophidia, suggesting that EGFP^A206K^ does not promote CTPS assembly. mCherry and miRFP670nano are also monomeric tags, just like EGFP^A206K^. However, hCTPS1-EGFP can form filament-shaped condensates without DON treatment (Appendix A). Since EGFP has a force to form dimer-like “sticky” features, and hCTPS1-EGFP molecules stick to each other in filament-shaped condensates, we refer to these filament-shaped hCTPS1-EGFP structures as “cytoophidium-like condensates” (to be distinguished from the term “cytoophidia”) (Figure 1C).

Are cytoophidia just condensates? If they are, the key cytoophidium-forming site CTPS-355H may provide a directional force of assembly, which should be important for filamentous condensate formation. For hCTPS1^H355A^-EGFP, the force of assembly is provided by dimeric EGFP, since CTPS-355H has been mutated to CTPS^H355A^. If either CTPS-355H or EGFP can provide force for condensate formation, we would expect that both hCTPS1^H355A^-EGFP and hCTPS1-EGFP^A206K^ can form condensates. Our results show that hCTPS1^H355A^-EGFP cannot form cytoophidium-like condensates, suggesting that CTPS-H355 is an essential site of connection rather than just there to provide a directional force (Figure 1H).

Therefore, our data argue against the idea that cytoophidia are condensates. Two factors appear to be required for assembling CTPS into cytoophidia. First, CTPS molecules are brought together by some forces of assembly. Second, CTPS tetramers need to be connected via CTPS-355H.

### 3.2. Cytoophidia Are Dynamic

In order to solve the problem of the physical phase of cytoophidia, we carried out FRAP assays to measure the dynamic features. We used hCTPS1^H355A^-mCherry as the control for complete diffusion, which recovered its intensity quickly after bleaching, so that we could not capture the difference before and after bleaching or measure its dynamic value. We used hCTPS1-EGFP cytoophidium-like condensates as static controls. The intensity recovered in cytoophidia was achieved significantly faster than in hCTPS1-EGFP cytoophidium-like condensates (Figure 2F,G). This means that the cytoophidium is not a condensate but a highly dynamic structure.

A low concentration of DON can induce cytoophidia, mimicking the stress of severe glutamine deprivation. We also measured the effects of time and concentration of DON treatment to determine whether the dynamic results were valid only in particular circumstances. Compared with these curves, the treatment time of DON had little effect on the kinetics (Figure 2E,G), while the concentration of DON had a significant effect on the kinetics of cytoophidia (Figure 2D,G). Due to the covalent bond between DON and CTPS, excessive DON may destroy the conformation of CTPS, thus damaging cells.

### 3.3. Cytoophidia Are Reticular

The cytoophidium is on a micrometric scale. A previous assumption was that the cytoophidium might be a bundle of CTPS filaments, similar to actin filaments. When a bleached ROI gradually recovered its intensity, no treadmill phenomenon was observed, which means that the assembly mechanism of cytoophidia is different from that of actin filaments. Moreover, recovery did not come from either side of the bleached ROI. The intensity of the entire ROI recovered steadily at the same speed.

If the cytoophidium is a bundle of CTPS filaments, how can the bleached ROI recover its intensity without a treadmill phenomenon (Figure 2H)? To solve this problem, we need to resolve the fine structure of cytoophidia. We performed super-resolution STED imaging which revealed the reticular structure of cytoophidia. In order to minimize artificial influences on the imaging results, we used miRFP670nano as a fluorescence tag to conduct STED imaging directly on live-cell samples, which gave a clear reticular structure (Figure 3A). We called it a reticular structure because small subunit filaments were interconnected, crossed and woven into the reticulation (Figure 3B).

At present, it is unclear whether the subunit filament is a CTPS filament, a bundle of CTPS filaments or some other form of CTPS. To maximize the resolution of the STED imaging (Figure 3B), all fluorescent tags were infrared-emitting. miRFP670nano is much smaller than other infrared fluorescent proteins, and it can minimize the impact of tag-space obstruction. We performed immunofluorescence staining to confirm the results and avoid the effects of spatial obstruction and overexpression. Cytoophidia with immunofluorescence dye Cy5 on hCTPS also showed the reticular structure.

We also performed immunofluorescence staining on SW480 cells. Cytoophidia in SW480 cells can be induced by glutamine deprivation and present reticular structures, which means reticular cytoophidia are common in different cell types and under glutamine metabolic stress conditions but cannot be treated artificially. Even DON is used on cells to mimic the stress of glutamine metabolic stress.

We used hCTPS1-EGFP as the structurally static control to represent the condensate assembled by physical forces. hCTPS1-EGFP cytoophidium-like condensates do not show the reticular characteristics. The cytoophidium-like condensates appear homogeneous, and their internal parts look the same. There are neither subunit filaments nor can space be observed inside the cytoophidium-like condensates. In summary, the structure of the cytoophidium is reticular, and the arrangements of CTPS assembled on cytoophidia are different from those of condensates or actin filaments.

The cytoophidium reticulation provides a structural basis for the localization of other enzymes, such as IMPDH (Figure 3C,D). Both CTPS and IMPDH are associated with glutamine and NH_3_ metabolism. In reticular cytoophidia, there may be dynamic interaction between CTPS, IMPDH and their substrates and the microenvironment. The reticular structure can also provide elasticity for the bending or twisting of cytoophidia (Appendix A). It was reported that cytoophidia are related to the regulation of IMPDH activity [37].

The reticular structure provides a structural basis for the FRAP results. No treadmill phenomenon has been observed. How can CTPS molecules in a free state replace those composed of the bleached ROIs? Based on the assumption of the reticular structure, the treadmill may occur in the subunit filaments rather than the whole cytoophidium. The assembly and disassembly of CTPS on subunit filaments may have a dynamic equilibrium, thus changing the CTPS molecules after bleaching. This may be a potential explanation for the dynamic equilibrium of large-scale FRAP.

Due to the limitations of STED and confocal devices, we could not achieve super-resolution, high-speed live-cell imaging and low phototoxicity for the cells. In order to verify the speculations, more advanced microscope technology is required. Dynamic equilibria of assembly and disassembly in subunit filaments may contribute to the metabolic regulation of reactions in the microenvironment.

In *Caulobacter crescentus,* a small amount of CTPS forms a bundle, while a large amount of CTPS forms a splayed structure [2]. A large amount of hCTPS transforms the morphologies of CTPS bundles into complex structures. Cytoophidia in *Drosophila* female germ cells also exhibit reticular characteristics [1]. Cytoophidia are regulated by the level of molecular crowding in a cell [38]. The formation and maintenance of the reticular cytoophidia may be related to molecular crowding.

### 3.4. CTPS Can Form Granules with Tentacles

While using live-cell imaging to capture CTPS-containing structures, we found interesting CTPS granules with tentacles. CTPS granules move in a random-walk mode, but the granules with tentacles move in a clear direction. We assume that these two entities are different forms of compartments with similar shapes. We use the term “tentacle” because the structure slowly stretches out of a granule and quickly retracts, just like the tentacle of an octopus or snail (Figure 4B). The tentacle may be extended to find something that can be connected. The tentacles, like bridges, connect different granules (Figure 4A).

If the granules are very small, the tentacles play a role in directional movement (Figure 4C). The granules extend the tentacles to the maximal length, and the granules move quickly from one side to the other along the tentacles, just like a slingshot. The only function of tentacles we know of is related to the directional movement of granules. We still do not know the function of tentacles as bridges. Moreover, it is unclear whether the granules with tentacles have membranes, their movements being similar to the movements of mitochondria and vesicles from the Golgi.

Interestingly, CTPS granules, tentacled granules and cytoophidia appeared under the same conditions, i.e., treatment with DON. They can even exist in the same cell, with potential for interactions and transformations. The tentacles are also capable of directional movements, just like cytoophidia. However, due to their tiny size, the tentacles may not share the same reticular structure as cytoophidia.

The structure of the tentacle may be similar to the subunit CTPS filament to obtain directional characteristics, or it may be in a liquid state in the membrane vesicle. The intensities of granules with tentacles are far lower than those of cytoophidia, and the tentacles exist in fewer cells than cytoophidia do. Due to their low intensities and small volumes, it is difficult to analyze the properties and fine structures of the tentacles. Compared with cytoophidia reacting to metabolism, it is not clear whether granules with tentacles are related to glutamine metabolism or the role of DON.

## 4. Materials and Methods

### 4.1. Cell Culture

The 293T and SW480 cells were cultured in DMEM (SH30022.01, Hyclone; Cytiva; 100 Results Way, Marlborough, MA USA 01752) supplemented with 10% FBS (04–001; Biological Industries; Kibbutz Beit-Haemek, 25115, Israel) in a humidified atmosphere containing 5% CO_2_ at 37 °C. All the commercial cell lines used in this article were purchased from the Shanghai Institutes for Biological Sciences, Chinese Academy of Sciences (Shanghai, China). They were originally purchased from ATCC. DON was dissolved in PBS and was added to the culture medium as described in individual experiments. DMEM without glutamine (C11960500BT, Gibco; Thermo Fisher Scientific; 168 Third Avenue, Waltham, MA, USA 02451) replaced DMEM 8 h before imaging.

### 4.2. Constructs and Transfection

The pLV-hCTPS1-EGFP over-expression vector was kindly provided by Dr. Zhe Sun from ShanghaiTech University. mCherry, miRFP670nano replaced EGFP and EGFP*^A206K^* was mutated back into EGFP using PCR and a Gibson Assembly System (NEB). Cell transfection was performed with PEI reagent (24765-1, Polysciences; Polysciences, Inc.; 400 Valley Road, Warrington, PA 18976, USA), according to the instructions provided by the manufacturer. The sequences of oligonucleotides used in this study are listed in Appendix A.

### 4.3. Immunoblotting

Cells were harvested in lysis buffer (containing 20 mM Tris, 150 mM NaCl and 1% Triton X-100; P0013J, Beyotime; Beyotime Biotechnology; Building 30, Songjiang Science and Technology Entrepreneurship Center, 1500 Lane, Xinfei Road, Songjiang District, Shanghai, China 201611). Undissolved cell fractions were separated by centrifugation at 12,000 rpm for 10 min at 4 °C, and the supernatants were boiled in SDS-PAGE loading buffer for 10 min. Proteins in total cell lysates were separated by SDS-PAGE and transferred to PVDF membranes. Membranes were blocked in 5% nonfat milk and incubated with the appropriate primary antibodies. Protein bands were visualized using horseradish peroxidase (HRP)-conjugated secondary antibodies with ECL reagent (34577, Thermo Fisher Scientific; 168 Third Avenue Waltham, MA, USA 02451).

### 4.4. Immunofluorescence

Cells were fixed with 4% paraformaldehyde added into media for 25 min. Then, the fixed cells were washed in 1xPBS 3 times. Samples were incubated with appropriate primary antibodies (rabbit anti-IMPDH2, Proteintech 12948-1-AP; rabbit anti-CTPS, Proteintech 15914-1-AP; Proteintech Group, Inc.; 5500 Pearl Street, Suite 400 Rosemont, IL 60018, USA) overnight at 4 °C and washed in PBS 3 times. Samples were incubated with Cy5-conjugated secondary antibodies (donkey anti-rabbit Cy5-conjugated antibody, Jackson 711-175-152; Jackson ImmunoResearch Inc.; 872 West Baltimore Pike, West Grove, PA 19390, USA) at room temperature for 1 h (in the dark) and washed with PBS 3 times after incubation. The mountant used for STED imaging (Figure 3A,C,D; Appendix A) was Prolong^TM^ Diamond Antifade (Invitrogen, P36965; Thermo Fisher Scientific; 168 Third Avenue, Waltham, MA, USA 02451). The mountant used for confocal imaging (Figure 1H) was HardSet Mounting Medium with DAPI (VECTASHIELD, H-1500; Vector Laboratories, Inc.; 6737 Mowry Ave, Newark, CA 94560, USA)

### 4.5. Microscopy

Images (Figure 1A,B, Figure 2C, Figure 4A,C, Appendix A) were acquired under 100× objectives with a confocal microscope (Nikon CSU-W1 SoRa). Confocal images and super-resolution images (Figure 3A,C,D and Appendix A) were acquired under 100× objectives with an STED confocal microscope (Leica TCS SP8 STED 3X). Images (Appendix A) were acquired under 63× objective with a Lattice SIM microscope (Zeiss Elyra 7) in wide-field mode. Confocal images (Figure 1H) were acquired under 63× objective with a confocal microscope (Zeiss LSM 980 Airyscan2).

### 4.6. Live Imaging

The 293T cells transfected with hCTPS1-mCherry and hCTPS1-miRFP670nano constructs were cultured on glass-bottom culture dishes (C8-1.5H-N, Cellvis; Vitro Scientific; Mountain View, CA 94039, USA) with medium and maintained at 37 °C when the live imaging was performed.

### 4.7. Image Analysis

Fluorescence images was analyzed with the software IMAGEJ (NIH, Bethesda, MD, USA). The ROIs of bleached regions for intensity measurement shown in Figure 2 were selected and measured manually with IMAGEJ. The FRAP curve unpaired *t*-test was analyzed with Graph Prism 8.4.0 (GraphPad Software, LLC; San Diego, CA, USA). The deconvolution of STED images shown in Figure 3 was analyzed by the lighting algorithm of the Leica LAS X. The 3D model shown in Figure 3 was analyzed with the Leica LAS X. The lengths of the tentacles shown in Figure 4 were measured manually with IMAGEJ. Quantity data were collected with Microsoft Excel.

### 4.8. Fluorescence-Activated Cell Sorting (FACS) Analysis

The flow cell analyzer used was an LSRFortessa X20 (BD). Flow cell data were analyzed with FlowJo 10.4 (FlowJo, LLC; Ashland, OR, USA) software. Data were collected with Microsoft Excel 16.61 (Microsoft Corporation; Redmond, WA, USA).

## 5. Conclusions

To sum up, the main purpose of this study is to understand the structure and arrangement of CTPS in CTPS filaments with near-atomic-resolution and micron-scale cytoophidia observed under confocal microscopy. We use dimeric EGFP tags as controls to provide aggregation viscosity and identified the connecting role of the CTPS-355H site. FRAP analysis shows that the cytoophidium is highly dynamic, while STED analysis reveals the reticular structure of cytoophidia.

According to the comparison with CTPS-EGFP cytoophidium-like condensates, the dynamic and reticular characteristics of cytoophidia are different from those of condensates (Figure 3B). Moreover, we find that the compartments of CTPS not only exist in the snake-shaped cytoophidia but also in the granules. CTPS granules move in different ways depending on whether they have tentacles. To understand the functions of CTPS granules with tentacles, further studies are required.

## Figures and Tables

**Figure 1 ijms-23-11698-f001:**
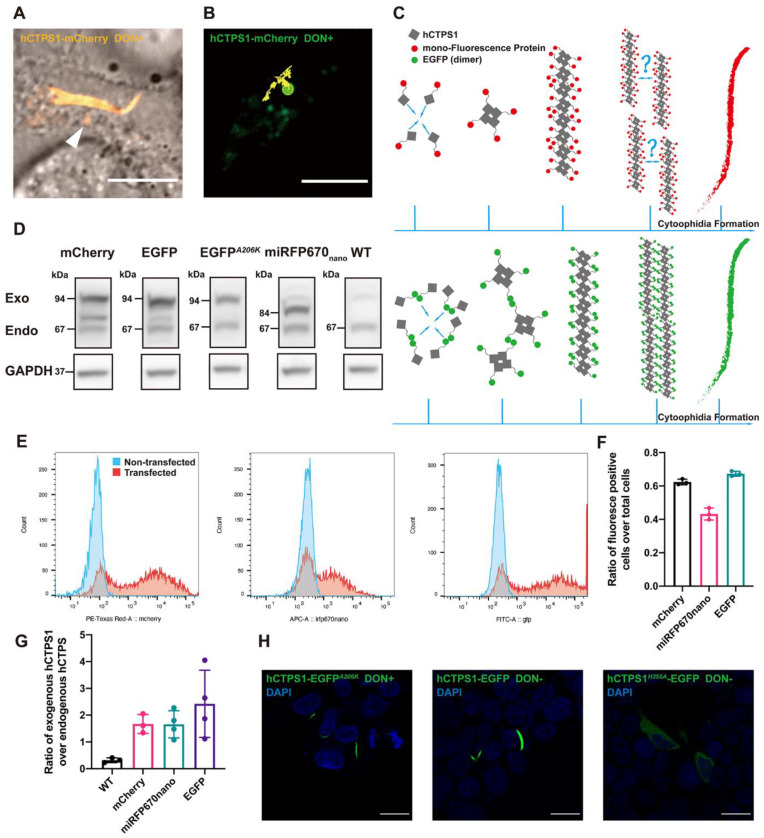
Assembly of CTPS filaments into cytoophidia. (**A**) hCTPS1 forms granules in the same cell that coexist with hCTPS1 cytoophidia. The arrowhead points to a granule. (**B**) The trajectory of hCTPS1 granules is a random walk. For the DON treatment, 20 μg/mL DON in PBS solution was added to fresh DMEM medium 8 to 25 h before live-cell imaging. (**C**) hCTPS1-EGFP forms cytoophidium-like condensates by simple force between EGFP. The arrangement from hCTPS1 filaments to hCTPS cytoophidia is the problem that needed to be solved. (**D**) The quantities of transfected over-expressed hCTPS in 293T cells were measured. (**E**,**F**) The transfection efficiencies were quantified. (**G**) Estimated ratio of exogenous hCTPS1 to endogenous hCTPS. (**H**) hCTPS-EGFPA206K cytoophidia can be induced by DON treatment. hCTPS1-EGFP can form cytoophidium-like condensates, which are wider and larger than hCTPS cytoophidia. hCTPS1H355A-EGFP cannot form cytoophidium-like condensates. For the DON treatment, 20 μg/mL DON (PBS solution) was added to fresh DMEM medium 8 h before fixation. Scale bars, 10 μm (**A**,**B**) and 20 μm (**H**).

**Figure 2 ijms-23-11698-f002:**
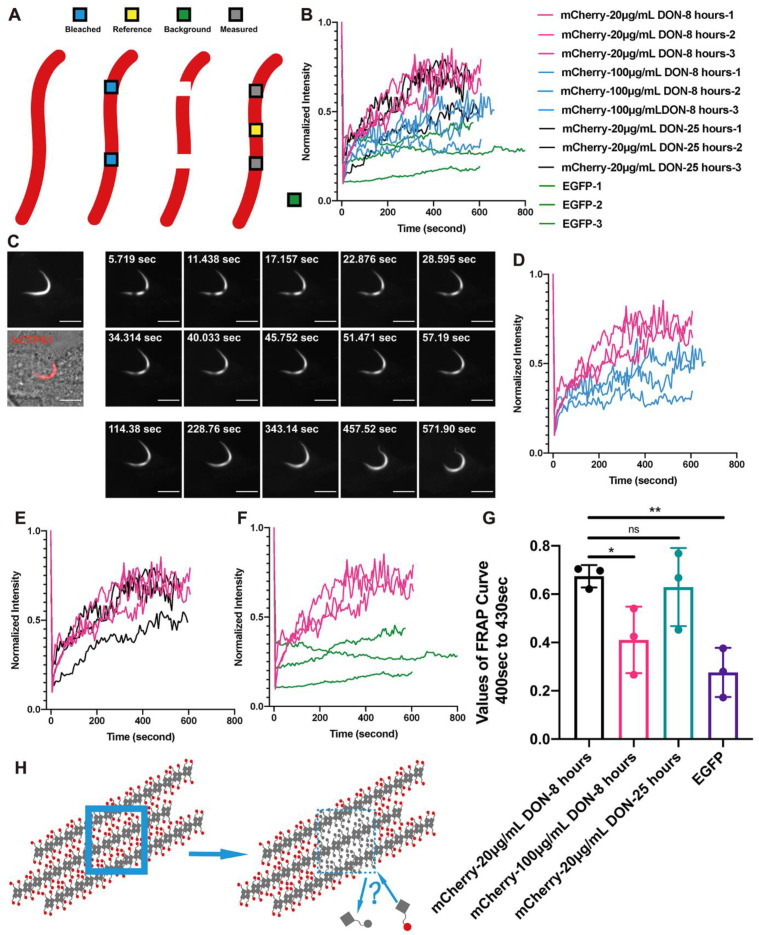
Dynamic equilibria of cytoophidia. (**A**) ROIs were used for bleaching, measurement and normalization of data. (**B**) Normalized intensity curves of FRAP results in different groups were merged. (**C**) Live-cell images of FRAP in hCTPS1-mCherry cytoophidia induced by 20 μg/mL DON for 8 h. (**D**) Comparison of FRAP curves for hCTPS1-mCherry between 20 μg/mL DON for 8 h and 100 μg/mL DON for 8 h. (**E**) Comparison of FRAP curves for hCTPS1-mCherry with 20 μg/mL DON for 8 h (pink curves) and 20 μg/mL DON for 25 h (black curves). (**F**) Comparison of FRAP curves for hCTPS1-mCherry cytoophidia (pink curves) induced by DON and hCTPS1-EGFP cytoophidium-like condensates (green curves). (**G**) Analysis of the level and speed of fluorescence recovery. *, *p*-value < 0.05; **, *p*-value < 0.01; ns, no significant difference. (**H**) The model of the structure of compact or condensed hCTPS cytoophidia does not fit the results of the FRAP images. A new model of structures is needed to explain the recovery from bleached fluorescence in the cytoophidia. The intensity of the FRAP ROI was normalized as $Normalized\;Intensity=\frac{I{\left(non-bleach\;ROI\right)}_{Pre-bleach,0}\;-\;I{\left(background\right)}_{0}\;}{I{\left(non-bleach\;ROI\right)}_{n}\;-\;I{\left(background\right)}_{n}}\;\times \;\left[I{\left(bleach\;ROI\right)}_{n}\;-\;I{\left(background\right)}_{n}\right]$. Scale bars, 10 μm (**C**).

**Figure 3 ijms-23-11698-f003:**
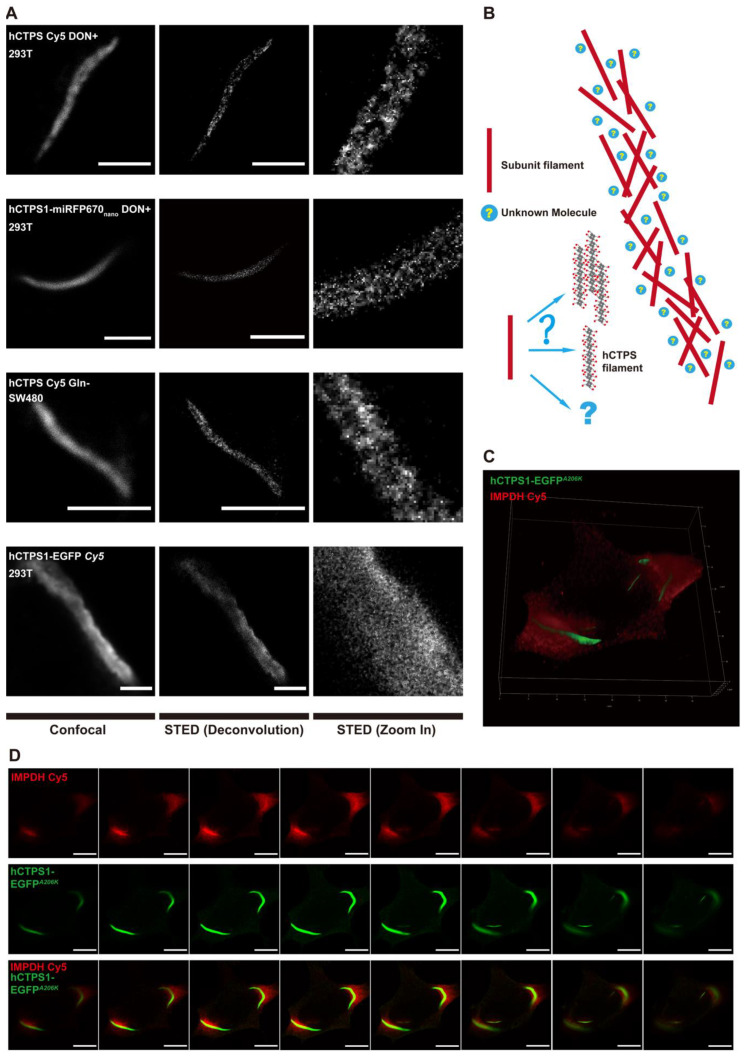
The reticular structure of the hCTPS1 cytoophidium and its localization with hIMPDH2. (**A**) Confocal STED with deconvolution and zoomed-in images of cytoophidia from the following groups: (1) hCTPS1/2 cytoophidia (Cy5 antibody-stained) induced by DON in fixed 293T cells, (2) hCTPS1-miRFP670nano cytoophidia induced by DON in live 293T cells, (3) hCTPS1/2 (Cy5-stained) cytoophidia induced by glutamine deprivation in fixed SW480 cells and (4) hCTPS1/2 (Cy5-stained)- cytoophidium-like condensates in live 293T (hCTPS1-EGFP-overexpression) cells. For the SW480 culture, DMEM without glutamine replaced DMEM 8 h before fixation. For the DON treatment, DON (PBS solution) was added to fresh DMEM medium 8 to 25 h before live-cell imaging or 8 h before being fixed. Scale bars, 3 μm. (**B**) The model of the arrangement of hCTPS filaments into cytoophidia. (**C**) In 293T cells, hCTPS1-EGFPA206K was localized adjacently with DON-induced IMPDH2 (Cy5-stained). (**D**) In each slice along the Z stacks, hCTPS1-EGFPA206K and IMPDH2 (Cy5-stained) were localized to each other. Scale bars, 10 μm.

**Figure 4 ijms-23-11698-f004:**
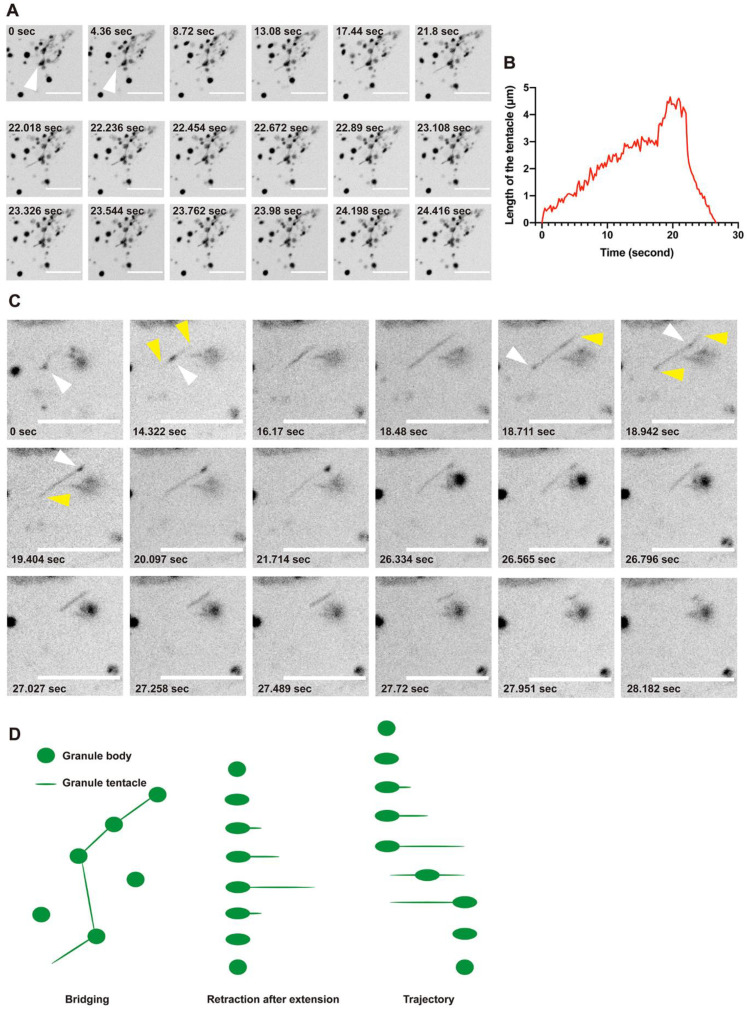
CTPS granules with tentacles. (**A**) Tentacles connect hCTPS1 granules. Tentacles extend and retract. Arrowheads indicate the tentacle. (**B**) Tentacles extend slowly and retract rapidly after reaching the maximum length. (**C**) hCTPS1 granules move from one side to the other along the tentacles. Yellow arrowheads indicate the tentacles; white arrowheads indicate the granules. (**D**) hCTPS1 granular tentacles have three different behaviors and characteristics, bridging, retraction after extension and the trajectory of hCTPS1 granule movement. For the DON treatment, DON (PBS solution) was added to fresh DMEM medium 8 to 25 h before live-cell imaging. Scale bars, 10 μm (**A**,**C**).

## Data Availability

Not applicable.

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
