# Peer review of "Super-Resolution Imaging Reveals Dynamic Reticular Cytoophidia"

_ijms, 2022, doi:10.3390/ijms231911698_

Round 1

Reviewer 1 Report

The work is devoted to the structural study of cytophidia in human cells using the FRAP (dynamics) and STED (super-resolution microscopy) methods. The study is interesting and potentially important, since without fully understanding the structure of an object, it is difficult to talk deeply about its function. The reviewer has no questions about the experimental part. All necessary controls are taken into account, the image quality is satisfactory.

The main claim to the manuscript:

In my opinion, the discussion of the work is extremely concise, and contains a very small number of sources. It is necessary to approach this section much more carefully and "chew" all the details for readers, with more examples and more serious comparative analysis. Particularly lacking is an analysis of the results concerning granules with the so-called tentacles. Please work on it. Without this, the article cannot be published in the IJMS (IF = 6.208).

Minor points:

Please check the manuscript carefully for errors and typos, as well as for English grammar (for examples, lines 50-51 – found, not find).

Lines 52, 140, etc. – “structured like a skeleton”, “appear as the skeleton of hCTPS cytoophidia” – I really don’t understand these phrases, please rephrase them (I don’t like the term “skeleton” in relation to these structures).

Author Response

Point 1: In my opinion, the discussion of the work is extremely concise, and contains a very small number of sources. It is necessary to approach this section much more carefully and "chew" all the details for readers, with more examples and more serious comparative analysis.

Response 1: We expanded the discussion substantially.

Point 2: Particularly lacking is an analysis of the results concerning granules with the so-called tentacles.

Response 2: We added description and analysis of the results concerning granules with tentacles.

Point 3: Please check the manuscript carefully for errors and typos, as well as for English grammar (for examples, lines 50-51 – found, not find).

Response 3: Revised.

Point 4: Lines 52, 140, etc. – “structured like a skeleton”, “appear as the skeleton of hCTPS cytoophidia” – I really don’t understand these phrases, please rephrase them (I don’t like the term “skeleton” in relation to these structures).

Response 4: We rephrase ‘skeleton’ to ‘reticulation’ and ‘skeletal’ to ‘reticular’.

Reviewer 2 Report

The paper "Super-resolution imaging reveals dynamic skeletal cytoophidia" by Yifan Fang et al devoted to the study of cytoophidia. The article is interesting. The experiments are justified and interpreted in most cases correctly. Figure 3B shows the arrangement model of https filaments into cytoophidia. However, it seems to me too bold to deduce the molecular structure of fibrils based on the method of light microscopy with a resolution of 50-70 nm. By the way, the authors have not proved that in their hands this method gives exactly such a resolution.

The second remark is that the author for the correspondence Ji-Long Liu has been dealing with the problem of cytoophidia formation for quite some time. However, they have not yet presented the ultrastructure of this organelle in cells. At least in the 2011 article I did not find this information. The structure of CTPS filaments have been solved by cryo-EM although only in vitro. However, it is necessary to confirm this in cell.

Finally, the discussion in the article looks very modest. What role this structure plays is poorly explained.

Author Response

Point 1: Figure 3B shows the arrangement model of https filaments into cytoophidia. However, it seems to me too bold to deduce the molecular structure of fibrils based on the method of light microscopy with a resolution of 50-70 nm.

Response 1: We updated the Figure 3B to a more rational model with various speculation options.

Point 2: By the way, the authors have not proved that in their hands this method gives exactly such a resolution.

Response 2: We added Figure S3A and description to illustrate how we estimate the resolution.

Point 3: Finally, the discussion in the article looks very modest.

Response 3: We expanded the discussion substantially.

Point 4: What role this structure plays is poorly explained.

Response 4: We described 3 potential roles in discussion.

Reviewer 3 Report

this report describes a detailed visualization of cellular structures termed "cytoophidia", that are formed under certain experimental conditions by the enzyme CTP synthase (CTPS).  Various CTPS-Fluorescent protein fusion constructs are prepared and analyzed using high resolution STED microscopy and FRAP assays.

Despite the extensive and abundant amount of data that was presented in this submission, I could not form a coherent image of exactly what the authors are proposing or postulating based on their results.  Part of the problem may be that the authors are insufficiently describing their experimental design (for example, in this study I "think" that hCTPS1-EGFP is being used as a structurally static control, that does not show FRAP, compared to for example hCTPS1-mCherry, owing to the fact that the former protein associates via interactions by EGFP, and not hCTPS1. Is this the case? If so, the authors need to describe their intentions in more detail.)

Related, the construct hCTPS1-EGFPA206K, is this construct a dynamic variant, and therefore represent an hCTPS1 analog in the cellular environment? If this is postulated to be the case, the authors need to provide evidence that this is indeed so.

 Confusing terminology, complex experimenal design, and a lack of sufficient explanations to tie all of the results together, all of these factors contribute, in this reviewer's opinion, to the overall disjointed impression of this manuscript.  The paper should be gone over in much detail to clarify the ideas proposed.

Author Response

Point 1: Despite the extensive and abundant amount of data that was presented in this submission, I could not form a coherent image of exactly what the authors are proposing or postulating based on their results.  Part of the problem may be that the authors are insufficiently describing their experimental design (for example, in this study I "think" that hCTPS1-EGFP is being used as a structurally static control, that does not show FRAP, compared to for example hCTPS1-mCherry, owing to the fact that the former protein associates via interactions by EGFP, and not hCTPS1. Is this the case? If so, the authors need to describe their intentions in more detail.)

Response 1: We added description of experiment designs in details in discussion.

Point 2: Related, the construct hCTPS1-EGFPA206K, is this construct a dynamic variant, and therefore represent an hCTPS1 analog in the cellular environment? If this is postulated to be the case, the authors need to provide evidence that this is indeed so.

Response 2: We added Figure S2E and description to illustrate EGFPA206K has no contribution to aggregation.

Point 3: Confusing terminology.

Response 3: We added explanation of these terminologies.

Point 4: Complex experimental design, and a lack of sufficient explanations to tie all of the results together, all of these factors contribute, in this reviewer's opinion, to the overall disjointed impression of this manuscript.  The paper should be gone over in much detail to clarify the ideas proposed.

Response 4: We added description of experiment designs in details and explanations to bring these results together with clear clues in discussion.

Round 2

Reviewer 1 Report

This manuscript is an improved and expanded work comparing with the previous version. Indeed, the authors have greatly widened the discussion by adding a number of references. The explanations now look more clear, the necessary details of the work have been discussed sufficiently. The reviewer thanks the authors for their work, but asks for a little more work to make the article look even better.

Firstly, there are still some questions about the style and grammar of the text (particularly, in the Discussion section). The reviewer recommends to check the manuscript carefully again to eliminate all shortcomings.

Secondly, thanks to the expanded text and careful explanations in the same Discussion section, there is now a lack of some kind of the Conclusion, where the main ideas of the text would be summarized.

If these two points are done, the manuscript can be accepted for publication.

Author Response

Point 1: Firstly, there are still some questions about the style and grammar of the text (particularly, in the Discussion section). The reviewer recommends to check the manuscript carefully again to eliminate all shortcomings.

Response 1: We have revised the manuscript, with extensive rewriting of the Discussion section.  For clarity, we have also added five subtitles in the Discussion section.

Point 2: Secondly, thanks to the expanded text and careful explanations in the same Discussion section, there is now a lack of some kind of the Conclusion, where the main ideas of the text would be summarized.

Response 2: We have added a Conclusion subtitle in the Discussion section.

Reviewer 3 Report

The authors have revised their initial submission with additional details and an experiment using the hCTPS1 mutant 355H, which perturbs oligomerization of hCTPS1. 

Although I recognize the authors' experimental efforts, I still fail to see in the revised submission any clear scientific hypotheses that are being established or proven. Part of the reason for this in my opinion is that the manuscript is too disjointed, describing many phenomena associated with hCTPS, but is not successful in establishing a correlation or meaningful relationship between any of them. To give an example, why do the authors insist on using the dimeric EGFP tag to visualize the association of hCTPS, when they clearly understand that usage of this dimer, which is able to induce association of the construct, will clearly complicate the interpretation of their experiments? There is no biological relationship between association of the hCTPS-EGFP construct and the actual formation of the cytoophidia in the cells, so I really do not understand why the authors persist in using data from this construct. They have numerous alternatives (the monomeric mutant, the mCherry label etc) so, what is the reasoning behind this persistence? Because quite frankly the results only add to the confusion when considering the biological relevance of the authors' results.

Other aspects of the paper only touch upon the surface of multiple phenomena, and do not probe further. What is the biological significance of the "tentacle" forms, for example. Are they formed under certain conditions? are they observable using the fluorescence constructs? Many questions remain after reading this submission. A clearer presentation in the main text, and an explanation of the author's intent in their rebuttal letter, would partially alleviate these impressions, but are not provided.

 Therefore, my overall impression is that the manuscript may still benefit greatly by clarifying exactly what the authors wish to state from their extensive experiments. There is very interesting data in these experiments, and I think that it would be unfortunate if the authors' ideas are not conveyed more clearly. 

Author Response

Point 1: Although I recognize the authors' experimental efforts, I still fail to see in the revised submission any clear scientific hypotheses that are being established or proven. Part of the reason for this in my opinion is that the manuscript is too disjointed, describing many phenomena associated with hCTPS, but is not successful in establishing a correlation or meaningful relationship between any of them.

Response 1: We have revised the manuscript, with extensive rewriting of the Discussion section.  For clarity, we have also added five subtitles in the Discussion section (details see below).

Point 2: To give an example, why do the authors insist on using the dimeric EGFP tag to visualize the association of hCTPS, when they clearly understand that usage of this dimer, which is able to induce association of the construct, will clearly complicate the interpretation of their experiments? There is no biological relationship between association of the hCTPS-EGFP construct and the actual formation of the cytoophidia in the cells, so I really do not understand why the authors persist in using data from this construct. They have numerous alternatives (the monomeric mutant, the mCherry label etc) so, what is the reasoning behind this persistence? Because quite frankly the results only add to the confusion when considering the biological relevance of the authors' results.

Response 2: The reason we insist to include dimeric EGFP in our experimental design is to take advantage of its ‘sticky’ feature. We wish to test the idea whether cytoophidia are just condensates. For condensate formation, a directional force between molecules is important. Dimeric EGFP (but not monomeric EGFPA205K) can provide directional force for condensate formation. In the Discussion section, we added the following sentences to explain the rationale -

“For hCTPS1H355A-EGFP, the directional force is provided by dimeric EGFP, since CTPS-355H has been mutated to CTPSH355A. If either CTPS-355H or EGFP can provide directional force for condensate formation, we would expect that both hCTPS1H355A-EGFP and hCTPS1-EGFPA206K can form condensates. Our results show that hCTPS1H355A-EGFP cannot form cytoophidium-like condensates, suggesting that CTPS-H355 is an essential site of connection rather than just providing a directional force (Figure 1H).”

Point 3: Other aspects of the paper only touch upon the surface of multiple phenomena, and do not probe further. What is the biological significance of the "tentacle" forms, for example. Are they formed under certain conditions? are they observable using the fluorescence constructs?

Response 3: We have now added a subtitle “CTPS can form granules with tentacles” in the Discussion section to summarize what we have learned about the tentacle. We also speculated potential functions and biological significance of the tentacle. In our experiment setting, we observed the tentacles under DON treatment. Yes, the tentacles are observable using the fluorescence constructs. 

Point 4: Therefore, my overall impression is that the manuscript may still benefit greatly by clarifying exactly what the authors wish to state from their extensive experiments. There is very interesting data in these experiments, and I think that it would be unfortunate if the authors' ideas are not conveyed more clearly. 

Response 4: Many thanks for the encouraging words. We have revised the manuscript, with extensive rewriting of the Discussion section.  For clarity, we have also added five subtitles in the Discussion section.

Round 3

Reviewer 3 Report

The revised version of the manuscript is much clearer, and the significance of the authors' results are highlighted.